# Blunt Chest Trauma in Polytraumatized Patients: Predictive Factors for Urgent Thoracotomy

**DOI:** 10.3390/jcm10173843

**Published:** 2021-08-27

**Authors:** Josef Stolberg-Stolberg, Jan Christoph Katthagen, Thomas Hillemeyer, Karsten Wiebe, Jeanette Koeppe, Michael J. Raschke

**Affiliations:** 1Department of Trauma-, Hand- and Reconstructive Surgery, Albert-Schweitzer-Campus 1, University Hospital Muenster, Building W1, 48149 Muenster, Germany; Christoph.Katthagen@ukmuenster.de (J.C.K.); michael.raschke@ukmuenster.de (M.J.R.); 2Department of Anesthesiology, Intensive Care, and Pain Medicine, Albert-Schweitzer-Campus 1, University Hospital Muenster, Building A1, 48149 Muenster, Germany; Thomas.Hillemeyer@ukmuenster.de; 3Section of Thoracic Surgery and Lung Transplantation, Department of Cardiothoracic Surgery, Albert-Schweitzer-Campus 1, University Hospital Muenster, Building A1, 48149 Muenster, Germany; Karsten.Wiebe@ukmuenster.de; 4Institute of Biostatistics and Clinical Research, University of Muenster, Schmeddingstrasse 56, 48149 Muenster, Germany; Jeanette.Koeppe@ukmuenster.de

**Keywords:** blunt chest trauma, polytrauma management, chest injury, thoracotomy

## Abstract

Purpose: Current guidelines on urgent thoracotomy of polytraumatized patients are based on data from perforating chest injuries. We aimed to identify predictive factors for urgent thoracotomy after chest-tube placement for blunt chest trauma in a civilian setting. Methods: Polytraumatized patients (Injury Severity Score ≥16) with blunt chest trauma, submitted to a level I trauma centre during a period of 12 years that received at least one chest tube were included. Trauma mechanism, chest-tube output, haemoglobin values, need for cellular blood products, coagulopathies, rib fracture pattern, thoracotomy, and mortality were retrospectively analysed. Results: 235 polytraumatized patients were included. Patients that received urgent thoracotomy (UT, *n* = 10) showed a higher mean chest-tube output within 24 h with a median (Mdn) of 3865 (IQR 2423–5156) mL compared to the group with no additional thoracic surgery (NT, *n* = 225) with Mdn 185 (IQR 50–463) mL (*p* < 0.001). The cut-off 24-h chest-tube output value for recommended thoracotomy was 1270 mL (ROC-Curve). UT showed an initial haemoglobin of Mdn 11.7 (IQR 9.2–14.3) g/dL and an INR value of Mdn 1.27 (IQR 1.11–1.69) as opposed to Mdn 12.3 (IQR 10–13.9) g/dL and Mdn 1.13 (IQR 1.05–1.34) in NT (haemoglobin: *p* = 0.786; INR: *p* = 0.215). There was an average number of 7.1(±3.4) rib fractures in UT and 6.7(±4.8) in NT (*p* = 0.649). Conclusions: Chest-tube output remains the single most important predictive factor for urgent thoracotomy also after blunt chest trauma. Patients with a chest-tube output of more than 1300 mL within 24 h after trauma should be considered for transfer to a level I trauma centre with standby thoracic surgery.

## 1. Introduction

Trauma is the leading cause of mortality and disability amongst young adults [1]. While neurologic injury is responsible for 60% of fatalities, exsanguination after blunt trauma has been described as major cause of preventable deaths [2,3]. Chest trauma ranks amongst the most important injuries in polytraumatized patients, with an incidence of 60% and a mortality of up to 25% [4,5]. Risk factors for poor outcome include a high injury severity score (ISS), multiple rib fractures, age over 65 years and injuries to lungs, heart or thoracic vessels [6,7]. Thus, scoring systems such as the Thoracic Trauma Severity Score, Pulmonary Contusion Score and Wagner Score have been developed for outcome prediction [8]. However, so far there is no clinical tool, risk score or recommendation available that evaluates the probability for the need of additional thoracic intervention after blunt chest trauma and chest-tube placement in polytraumatized patients.

Upon admission of a polytraumatized patient to the hospital, three time intervals can be differentiated: firstly, emergency-department thoracotomy is used for management of patients in extremis e.g., for haemorrhage control or relief of cardiac tamponade [9]. After stabilization the patient should be brought to the operating room for definite care. However, survival rates range only from 7–12% [10,11]. Secondly, urgent thoracotomy (UT) within 48 h is recommend in cases of continuous massive air leak, air embolism and if the initial blood loss after chest-tube placement measures >1500 mL or >250 mL/h over 4 consecutive hours any time after tube placement, according to the German S3 guideline on treatment of polytrauma and severe injuries [12]. US recommendations advocate UT, if blood loss exceeds 1500 mL or >20 mL/kg body weight after chest-tube placement, more than 200 mL/h over 3 consecutive hours, persistent bleeding at a rate > 7 mL/kg/h or refractory shock [13,14,15,16,17]. However, this recommendation is based on data from mostly penetrating bullet or fragment injuries [18]. Some authors suggest 1000 mL initial chest tube drainage in cases of pulmonary parenchymal bleeding, blunt chest trauma, delayed presentation or coagulopathy without evidence in literature [9]. Thirdly, the concept of delayed thoracotomy after 48 h is applied if more life-threatening injuries such as intracranial or abdominal bleedings have to be addressed first [19]. Summarizing, current guidelines on thoracotomy focus on emergency-department thoracotomy after penetrating injuries. However, indication for thoracotomy after blunt trauma, initial stabilization and hospital admission might vary considerably. Furthermore, the role of coagulopathies, initial haemoglobin and analysis of rib fractures might give additional information and have not been analyzed as predictive factors for thoracotomy so far. The aim of this study is to evaluate if current guidelines for thoracotomy also apply for blunt chest trauma and to analyze if other clinical values might also be reliable predictive factors.

## 2. Patients and Methods

### 2.1. Study Design

During the time interval of 12 years (2009–2020), all patients that received a chest tube in one level I trauma centre were included. They have been identified using the German Diagnosis Related Group-System. The clinical information system was scanned for the operation and procedure codes 8–144 and 5.340.0 (therapeutic drainage of the pleural cavity and surgical drainage of the chest wall). Retrospective data collection has been approved by the local ethics committee (ethics committee of the Medical Association of Westfalen-Lippe, no: 2018-015-f-S). Excluded were patients without traumatic chest injury, pretreatment for longer than 24 h in other hospitals, penetrating chest injury, incomplete data and false coding. The data have then been screened for ISS and thoracic abbreviated injury scale (AIS). All patients with ISS < 16 and thoracic AIS < 3 have been excluded (Figure 1). Data were collected using the electronic patient record system according to patient’s age, sex, trauma mechanism, ISS-, AIS- score, number and pattern of rib fractures, initial haemoglobin value, concomitant injuries, coagulopathy, thoracic surgery, chest-tube output, need of blood products and mortality (Table 1).

### 2.2. Outcome Measures and Definitions

The study outcome was defined as need for UT additional to chest-tube placement. Secondary outcomes measured chest-tube output during the first 24 h after trauma, initial haemoglobin value, need for cellular blood products, initial International Normalized Ratio (INR) value, number and pattern of rib fractures and mortality. According to the international definition, polytrauma was defined as ISS ≥ 16. In this study, flail chest refers to neighbouring segmental rib fractures ≥3 [20,21]. 

### 2.3. Statistics

Statistical analyses were performed using IBM SPSS^®^ Statistics 27 (IBM Corporation, Somers, NY, USA). Group differences were analyzed with a Chi-square test. Non-parametric Mann–Whitney–U test was performed to account for statistical noticeable differences of chest-tube output, coagulopathy, initial haemoglobin, need for blood products and number of rib fractures. The predictive value for thoracotomy was calculated using a receiver operating characteristic (ROC) curve model. All analyses were explorative and *p*-values are regarded noticeable, if *p* ≤ 0.05 without an adjustment for multiple testing.

## 3. Results

### 3.1. Study Population and Follow-Up

During the 12-year period, 383 patients were counted. After applying the inclusion criteria, 235 patients remained of whom 181 (77.0%) were men and 54 (23.0%) women. Their age ranged from 18 to 94 years with a mean of 52 years. Emergency treatment followed S3 guidelines on treatment of polytrauma and severe injuries, as stated above [12,22]. Regarding mortality, 216 (91.9%) patients survived after initial emergency stabilization and 198 (84.3%) of them survived during acute hospital stay. Both, the UT group and the group without additional thoracotomy (NT) were comparable. There was no significant difference in age, sex, ISS and AIS (Table 1).

### 3.2. Trauma Mechanism

Reason for chest trauma is known for all 235 patients. According to the registry of the German Trauma Society, mechanisms were specified into 10 categories: approximately one third of our patients had a fall, of which *n* = 25 (10.6%) fell < 3 m and *n* = 57 (24.3%) > 3 m height. Traffic accidents accounted for more than half of our patients; vehicle occupants *n* = 65 (27.7%), motorcyclists *n* = 27 (11.5%), cyclists *n* = 21 (8.9%), pedestrians hit by a vehicle *n* = 14 (6.0%) and category “others” *n* = 4 (1.7%) that were all hit by a train were counted. The category miscellaneous is divided into strike injury *n* = 7 (3.0%), and “others” *n* = 15 (6.4%), which comprise mainly accidents with agricultural machinery *n* = 4 (1.7%) and *n* = 11 (4.7%) cases of crush injuries (Figure 2).

### 3.3. Chest-tube output

Chest-tube output of the first 24 h has been recorded for 223 patients. The remaining 12 patients died shortly after arrival in the hospital, 5 due to severe head injury and 7 under cardiopulmonary resuscitation. Ten patients underwent UT due to internal bleeding within the first 48 h after trauma, with a median chest-tube output of median (Mdn) 3865.0 (interquartile range (IQR) 2422.5–5156.3) mL (max. 7920.0 mL; min. 1290.0 mL). Patients of NT (*n* = 213) had a median chest-tube output of Mdn 185.0 (IQR 50.0–462.5 mL) (max. 2400.0 mL; min. 0.0 mL) during the first day (Figure 3). Comparing UT with NT, there was a statistically noticeable difference in chest-tube output during the first 24 h after trauma (*p* < 0.001). There were 10 patients in NT that showed a chest-tube output above 1300 mL with a Mdn of 1770.0 (IQR 1487.5–2075.0) mL (max. 2400.0 mL; min. 1370.0 mL), which was statistically notably different to UT (*p* = 0.002). 

The ROC-curve displays the trade-off between sensitivity and specificity over a continuous range. The test optimum is nearest to the left-upper corner defined by sensitivity and 1-specificity [23]. Analysing the ROC-curve diagram the area under the curve was 0.991. A sensitivity of 100% and specificity of 95.3% at 1270 mL chest-tube output with a Euclidean distance of 0.047 was calculated (Figure 4). Delayed thoracotomy was necessary in three patients for an atypical wedge resection, because of air leakage (two times) and one purulent infection after 11, 24 and 45 days. For all of them, surgery was started with video-assisted thoracoscopic surgery (VATS) and converted to open thoracotomy. These three patients showed a chest-tube output of 710, 0 and 940 mL during the first 24 h. The cut-off value for UT was determined at 1300 mL.

### 3.4. Initial Haemoglobin and Need for Cellular Blood Products

The median initial haemoglobin value of UT was Mdn 11.7 (IQR 9.2–14.3) g/dL as opposed to Mdn 12.3 (IQR 10–13.9) g/dL of patients that needed no or delayed thoracotomy (*p* = 0.786). Within the first three hours of hospital treatment, UT needed averagely 2.4 (±4) units of concentrated red cells (CRC), 2.8 (±6.6) units of fresh frozen plasma (FFP), 0.4 (±1.3) units of platelet concentrates (PLT) as opposed to 1.1 (±2.9) CRCs, 1 (±3.6) FFPs and 0.1 (±0.5) PLTs of NT (Figure 5a, CRC: *p* = 0.222; FFP: *p* = 0.292; PLT: *p* = 0.331). Within a 24 h interval, 7.2 (±12.3) CRCs, 7.5 (±13.2) FFPs, 1.6 (±2.7) PLTs were needed in UT versus 2.8 (±6.4) CRCs, 3.2 (±7.9) FFPs, 0.4 (±1.2) PLTs in NT (CRC: *p* = 0.293; FFP: *p* = 0.149; PLT: *p* = 0.120) (Figure 5b). 

### 3.5. Coagulopathies

Initial coagulation status is recorded with the INR value (standard value 0.8–1.2). UT showed an initial INR value of Mdn 1.27(IQR 1.11–1.69) as opposed to NT with a median value of Mdn 1.13 (IQR 1.05–1.34) (*p* = 0.215).

### 3.6. Rib Fractures

Based on radiographic imaging, the number of rib fractures could be assessed in 234 patients. Concomitant injuries of the shoulder girdle as well as abdominal and pelvic injuries are listed in Table 1. One patient died under cardiopulmonary resuscitation before X-ray imaging. Of 234 bluntly injured patients, 206 showed rib fractures. Unilateral thoracic trauma (*n* = 156) fractured averagely 4.7 (±3.5) ribs on one side. Bilateral rib fractures (*n* = 78) had on average 10.7 (±4.4) on both, 7.2 (±2.8) on the more seriously and 3.5 (±2.3) fractures on the less seriously injured side. Of these, 184 (78.6%) of the patients were counted with serial rib fractures ≥3 and 144 (61.5%) ≥5 rib fractures. Forty-one (17.5%) of the patients suffered a sternum fracture, while 21 patients received rib plating, with the diagnosis of flail chest and respiratory insufficiency. Neither the total number of rib fractures, nor the number of rib fractures on the more seriously injured side, nor the presence of a sternum fracture was a predictive value for thoracotomy (total number: *p* = 0.649; more seriously injured side: *p* = 0.705; sternum: *p* = 0.157). There was no correlation between total number of rib fractures (Figure 6b) (Pearson r = 0.153, *p* = 0.023) or the number of rib fractures on the more seriously injured side (Figure 6a) (Pearson r = 0.100, *p* = 0.135) with chest-tube output within the first 24 h.

### 3.7. Mortality

Of 235 patients, 19 died during the first 24 h (10 × craniocerebral trauma, 9 × during CPR; ISS = 51.3 ± 14.9) and 18 within the acute inpatient hospital stay (7 × craniocerebral trauma, 11 × multiple organ failure; ISS = 39.6 ± 15.1). Patients that died during the hospital stay (>24 h) did not show noticeably higher chest-tube output within the first 24 h (Mdn 310 IQR 42.5–1088 mL) as compared to patients that survived (Mdn 183 IQR 50–468 mL) (*p* = 0.220). Patients that died during the first 24 h after submission had a statistically notably higher chest-tube output with Mdn 1450 mL (IQR 550–2000 mL) (*p* = 0.001). Patients that died had statistically and notably higher initial INR and lower initial haemoglobin values, with Mdn 1.84 (IQR 1.25–2.31), compared with Mdn 1.11 (IQR 1.04–1.29) (*p* < 0.001) and Mdn 9.7 (IQR 8.5–12.8) g/dL compared with Mdn 12.5 (IQR 10.4–14) g/dL (*p* = 0.001). There was no association between the number of rib fractures and mortality.

## 4. Discussion

The most important finding of this study was that a chest-tube output above 1300 mL/24 h remains the single most predictive factor for UT, as well as after blunt chest trauma. As opposed to penetrating chest trauma, there was no trauma mechanism associated with higher rates of UT. Furthermore, initial haemoglobin value, needed for cellular blood products, coagulopathies or rib fracture patterns showed no association with UT.

The measurement of chest-tube output has traditionally been the gold standard to decide for thoracotomy and the current S3 guideline still recommends surgical intervention in cases of initial blood loss after chest-tube placement >1500 mL or >250 mL/h over 4 consecutive hours [12,22]. However, studies vary significantly and mainly comprise penetrating chest trauma and battlefield injuries, therefore it is not clear if similar numbers can be applied for blunt chest trauma in a civil environment [24]. Mizushima et al. reported a cut-off for urgent thoracotomy of 1-h chest-tube output of 404 mL after blunt trauma analysing 117 patients [25]. Karmy-Jones et al. analyzed 36 patients with an average chest-tube output of 2220 ± 1235 mL before intervention. They recommend thoracotomy in cases of chest tube drainage of 1500 mL/24 h based on a mortality calculation [26]. Data of our study shows nearly twice the median volume with 3865 (IQR 2423–5156) mL and similar cut-off value of 1300 mL. However, it has to be outlined that none of the patients of our study suffered an injury of the heart or aorta, which might have caused an increased chest-tube output. Furthermore, ten patients of NT showed a chest-tube output above 1300 mL. Although we calculated a median of Mdn 1770.0 (IQR 1487.5–2075.0) mL, which was, statistically, notably less in comparison with UT, due to the retrospective assessment of this study, it is unknown if these patients might have profited from UT. However, current practice and expert opinion classify them as high risk for urgent or delayed thoracotomy, justifying a cut-off value of 1300 mL [9].

The secondary aim of this study was to identify further predictive factors for UT in polytraumatized patients. The current German S3 guideline on treatment of polytrauma and severe injuries is based on military experience from the 1970s with mainly bullet and fragment wounds and only 5% blunt trauma [18]. However, demographics and trauma mechanisms in developed countries with strict gun-control laws might be distinctly different [27]. Furthermore, medical technological progress, such as the introduction of VATS and rib plating, extend the indication for interventions of the chest. Thus, clinical decision-making and parameters in a civilian setting need to be re-evaluated. The aim of this study was the early identification of patients that profit from an additional intervention, as this could help to improve patient outcome and decrease mortality [28].

First, the trauma mechanism of polytraumatized patients is known to the admitting hospital pre-arrival in most cases, and the literature shows that early consultation of a cardiothoracic surgeon might increase survival rates of patients with blunt chest trauma [29,30]. Consistent with other authors, falls >3 m, as well as motor-vehicle accidents, are the prevalent injury mechanism of this study [31]. We found that 4.3% of the patients underwent secondary interventions of the chest, which is also similar to literature with 2.7–6% [32]. Although only a small percentage of patients in this study finally needed UT, our data indicates that polytraumatized patients with serious blunt chest trauma should be considered for direct transport to a level I trauma centre. In everyday clinical practice, the trauma surgeon on duty should explicitly ask the admitting emergency doctor for signs of a severe chest trauma and precociously consult a cardiothoracic surgeon.

Secondly, early coagulopathy can contribute to uncontrolled haemorrhage, morbidity and mortality [26]. Although persistent chest bleeding triggers thoracotomy, the role of blood coagulation after chest trauma has not been studied so far. We hypothesized that an initial INR value above 1.2 increases the probability of intrathoracic bleeding and surgical procedures to achieve haemostasis, which could not be confirmed. However, consistent with literature, a high initial INR value was associated with an increased mortality [27]. Furthermore, persistent blood transfusions to maintain hemodynamic stability has been suggested to trigger early operative management of a hematothorax [28]. However, the practical use has not been proven so far. Data from this study suggest that the number of CRC units cannot be used to predict thoracotomy. However, CRC transfusion is the consequence of continuous bleeding, and UT patients in this study would have needed further CRCs without early intervention. Hence, for clinical practice the initial INR has a higher prognostic value for UT than CRCs which are administered int the course of treatment.

Third, rib fractures are associated with high morbidity, mortality and are indicators for concomitant organ injury [21,33]. While it is known that specific mechanisms of injury cause corresponding rib fracture patterns, to our knowledge, the link to chest-tube output and thoracotomy has not been studied [34]. Data of this study could not show evidence for rib fracture patterns that are associated with high chest-tube output or urgent thoracotomy—nor was there a correlation to rib plating. The reasons for this might be given by differences in trauma mechanism and intraparenchymal pulmonary haemorrhage that could not be collected by the chest tube. Furthermore, rib plating is a relatively new technique that was applied within this study in the year 2013 for the first time [35]. Hence, in-practice evaluation of rib and sternal fractures, lung contusions and clinical signs, such as paradoxical breathing, needs to be done individually for each patient. Although there seems to be no correlation to UT, other interventions such as rib plating, invasive ventilation and prolong observation need to be considered.

Summarizing for clinical practice, chest-tube output can easily be measured and is an important element when evaluating the need for UT. However, by itself it is not sufficient to base a decision in favour or against urgent thoracotomy. Other factors, such as radiographic imaging, hemodynamic and coagulation status, have to be considered additionally [25]. Furthermore, it has to be argued that there might occur chest-tube drainage fluctuations or even inadequate drainage, which can cause further delay and increased mortality [26]. Consequently, patients with blunt chest trauma must be re-evaluated frequently and monitored. The final decision in favour or against urgent thoracotomy remains multifaceted and is based on many clinical factors. The total chest-tube output is one of them and can easily be measured. Recommending early transfer to a level I trauma centre might be a strategy to re-assess the patient by a thoracic surgeon with more experience, interpreting the clinical factors in order to reduce morbidity and mortality.

Limitations of this study include the small number of UT, the retrospective design and data having been collected from a single institution. Furthermore, within the study period, VATS and rib plating have been used with increasing frequency in trauma surgery due to technological progress. Another limiting factor is the heterogeneity of injury mechanisms and injury patterns in polytraumatized patients, which make it necessary to evaluate each patient individually. Thus, the measurement of the chest-tube output can only be one information source amongst others to make a clinical treatment decision.

## 5. Conclusions

Summarizing, chest-tube output remains the single most important predictive factor for thoracotomy in a civilian trauma setting. Patients that show a chest-tube output above 1300 mL/24 h after chest-tube placement should be considered for transfer to a trauma centre and presentation to a thoracic surgeon.

## Figures and Tables

**Figure 1 jcm-10-03843-f001:**
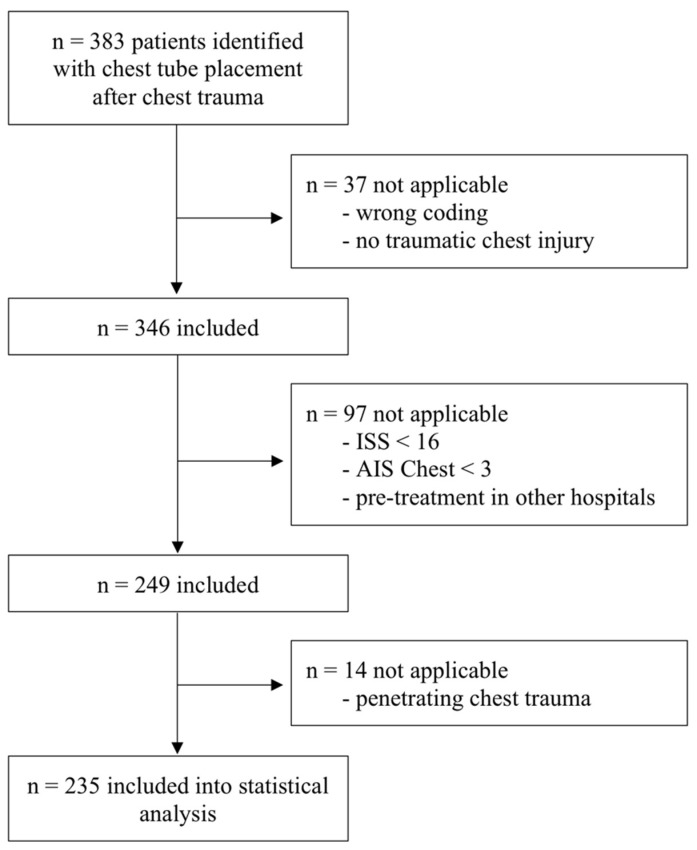
Flow diagram summarizing patient selection.

**Figure 2 jcm-10-03843-f002:**
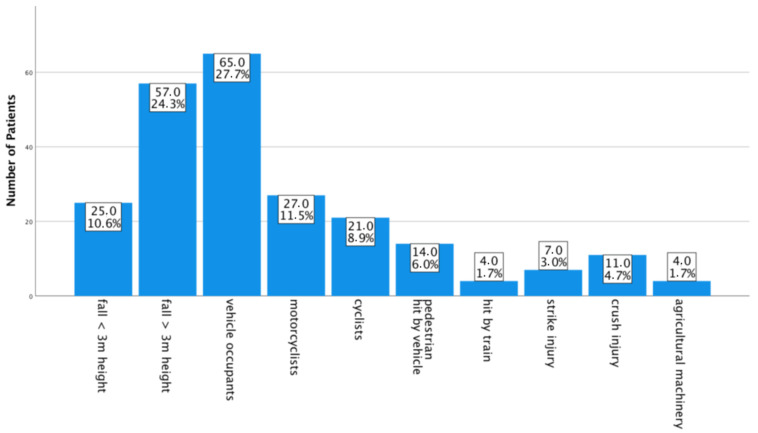
Chest trauma mechanism in total numbers and percentage.

**Figure 3 jcm-10-03843-f003:**
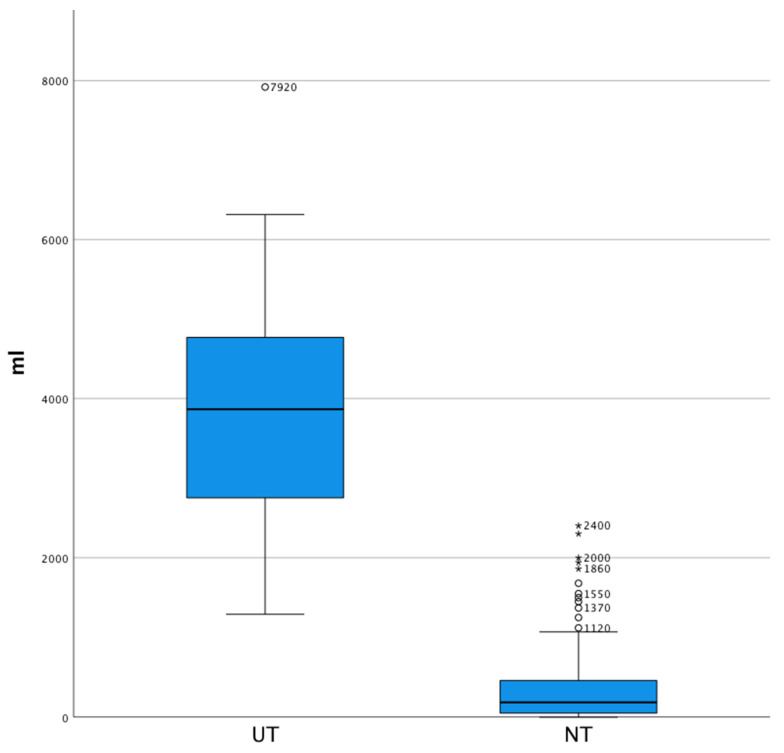
Boxplot diagrams comparing chest-tube output of UT and NT (*p* < 0.001). ^o^ indicates datapoints within 1.5–3 IQR, * above 3 IQR.

**Figure 4 jcm-10-03843-f004:**
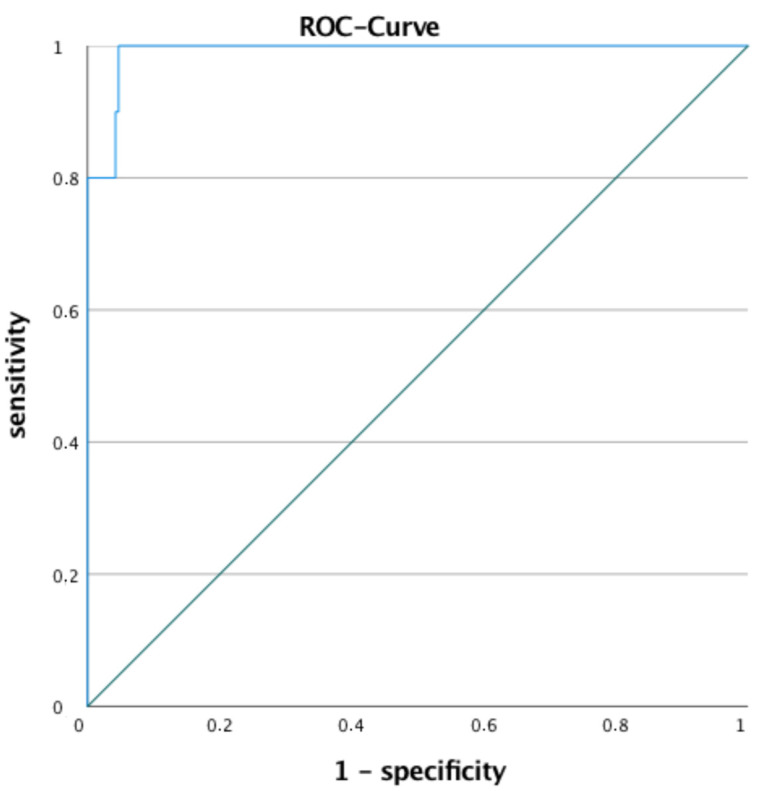
ROC-curve diagram on chest-tube output and need for UT shows a sensitivity of 100% and specificity of 95.3% at 1270 mL chest-tube output with an Euclidean distance of 0.047. The area under the curve is 0.991.

**Figure 5 jcm-10-03843-f005:**
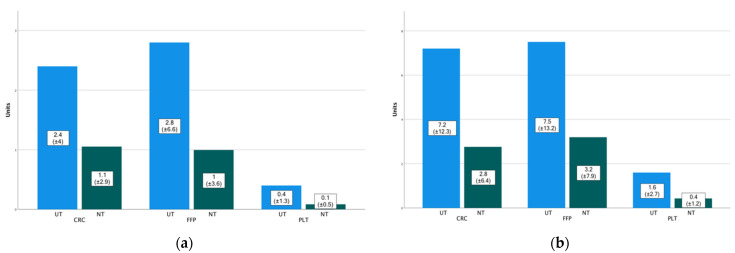
Bar diagram comparing the need for concentrated red cells, fresh frozen plasma and platelet concentrates between UT and NT within the first 3 h (**a**) and 24 h (**b**) after admission.

**Figure 6 jcm-10-03843-f006:**
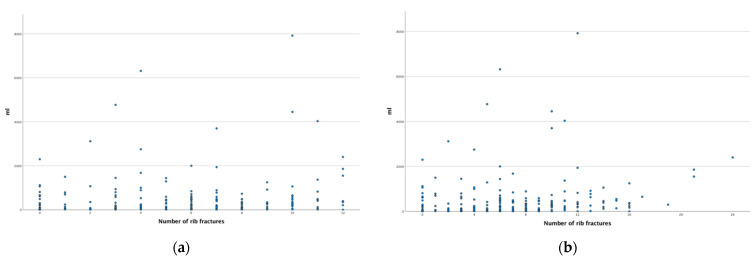
Scatter plot showing the relationship between the highest number of rib fractures on one side (**a**) and both sides (**b**) on one side with chest-tube output in mL.

**Table 1 jcm-10-03843-t001:** Baseline characteristics of study population.

	Total	No Urgent Thoracotomy	Urgent Thoracotomy
Number of patients	235	225 (95.7%)	10 (4.3%)
Sex			
male	181 (77.0%)	173 (76.9%)	8 (80.0%)
female	54 (23.0%)	52 (23.1%)	2 (20.0%)
age	52.0 (±20.1)	51.9 (±19.9)	55.1 (±24.9)
height in cm	178.3 (±9.1)	178.3 (±9.2)	179.0 (±7.1)
weight in kg	82.1 (±15.0)	82.3 (±15.1)	78.6 (±13.4)
BMI	25.9 (±4.0)	25.9 (±3.9)	25.0 (±4.7)
Death during the first 24 h?			
yes	19 (8.1%)	19 (8.4%)	0 (0.0%)
no	216 (91.9%)	206 (91.6%)	10 (100.0%)
Death during hospital stay?			
yes	37 (15.7%)	35 (15.6%)	2 (20.0%)
no	198 (84.3%)	190 (84.4%)	8 (80.0%)
ISS	35.4 (±13.7)	35.2 (±13.4)	39.7 (±18.5)
AIS region head	2.0 (±2.0)	2.0 (±2.0)	1.3 (±1.8)
AIS region face	0.7 (±1.2)	0.7 (±1.2)	0.2 (±0.4)
AIS region chest	3.6 (±0.7)	3.6 (±0.7)	4.2 (±0.9)
AIS region abdomen	2.2 (±1.8)	2.2 (±1.8)	3.2 (±1.4)
AIS region extremity and pelvis	2.0 (±1.5)	2.0 (±1.5)	2.2 (±1.5)
AIS region general/soft tissue	0.5 (±0.7)	0.5 (±0.7)	0.5 (±0.5)
ASA-Score	2.6 (±0.9)	2.6 (±0.9)	2.7 (±1.0)
Abdominal hemorrhage	51 (21.7%)	48 (21.3%)	3 (30.0%)
Pelvis Fracture AO Trauma			
total	64 (27.2%)	62 (27.5%)	2 (20.0%)
B	24 (10.2%)	23 (10.2%)	1 (10.0%)
C	40 (17.0%)	39 (17.3%)	1 (10.0%)
Scapula Fracture			
unilateral	48 (20.4%)	48 (21.4%)	0 (0.0%)
bilateral	2 (0.9%)	2 (0.9%)	0 (0.0%)
Clavicula Fracture			
unilateral	48 (20.4%)	47 (21.0%)	1 (10.0%)
bilateral	3 (1.3%)	3 (1.3%)	0 (0.0%)
AC Joint Disruption			
unilateral	7 (3.0%)	7 (3.1%)	0 (0.0%)
bilateral	0 (0.0%)	0 (0.0%)	0 (0.0%)
arterial hypertension	55 (23.4%)	53 (23.6%)	2 (20%)
atrial fibrillation/flutter	15 (6.4%)	15 (6.7%)	0 (0.0%)
atherosclerosis	10 (4.3%)	10 (4.4%)	0 (0.0%)
heart failure	9 (3.8%)	8 (3.6%)	1 (10.0%)
peripheral arterial disease	6 (2.6%)	6 (2.7%)	0 (0.0%)
obstructive pulmonary disease	15 (6.4%)	15 (6.7%)	0 (0.0%)
diabetes mellitus	13 (5.5%)	12 (5.3%)	1 (10.0%)
renal insufficiency	7 (3.0%)	6 (2.7%)	1 (10.0%)
hepatic insufficiency	2 (0.9%)	2 (0.9%)	0 (0.0%)
gastrointestinal tract disease	3 (1.3%)	3 (1.3%)	0 (0.0%)
hypothyroidism	9 (3.8%)	9 (4.0%)	0 (0.0%)
hypercholesterolemia	10 (4.3%)	10 (4.4%)	0 (0.0%)
substance abuse	4 (1.7%)	4 (1.8%)	0 (0.0%)
epilepsy	3 (1.3%)	3 (1.3%)	0 (0.0%)
paranoid schizophrenia	2 (0.9%)	2 (0.9%)	0 (0.0%)
dementia	3 (1.3%)	2 (0.9%)	1 (10.0%)
rheumatoid arthritis	2 (0.9%)	2 (0.9%)	0 (0.0%)

## Data Availability

Upon contact to the corresponding author.

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
