# Peer review of "Blunt Chest Trauma in Polytraumatized Patients: Predictive Factors for Urgent Thoracotomy"

_jcm, 2021, doi:10.3390/jcm10173843_

Round 1
Reviewer 1 Report
The authors evaluated predictive factors for urgent thoracotomy after chest tube placement for blunt chest trauma in a civilian setting. They concluded that chest tube output remains the single most important predictive factor for urgent thoracotomy also after blunt chest trauma. Patients with a chest tube output of more than 1300 ml within 24 h after trauma should be considered for transfer to a level I trauma center with standby thoracic surgery.
Overall, well-designed study, methodology reproducible but need to be improved in some basic information. Discussion should be improved, as well.
However, I have several suggestions and objections to improve the study:
- Please provide a full title for each abbreviation mentioned in the abstract. (IQR - should be stated at the place where this abbreviation is first used).
- Please provide clear inclusion / exclusion criteria in methodology.
- Please provide more baseline characteristics of the study population such as weight, height, BMI, comorbities…
- Please provide exact p-values instead of ‘’p>0.05’’ or ‘’p<0.05’’ through the results section.
- The authors stated that the predictive value for thoracotomy was calculated using a ROC curve mode. I do not see clear results of ROC analysis in the results section. Please can you provide area under the curve (AUC), sensitivity, specificity as well as ROC-curve diagram?
- Discussion is poorly designed. Discussion section needs to be re-written/re-arranged. Do not present a review of literature in this section. Do not discuss your findings piecemeal. Focus on results from the main objectives of the study. Write in four sequential paragraphs (without headings); (i) summary (not data) of findings from present study; (ii) logical and coherent comparison with existing literature with focus of comparison on main objective(s); (iii) limitations of the study and (iv) Implications for practice/policy/research with a concluding statement.
- Limitations of the study should be clearly stated at the end of discussion.
Author Response
Reviewer 1:
The authors evaluated predictive factors for urgent thoracotomy after chest tube placement for blunt chest trauma in a civilian setting. They concluded that chest tube output remains the single most important predictive factor for urgent thoracotomy also after blunt chest trauma. Patients with a chest tube output of more than 1300 ml within 24 h after trauma should be considered for transfer to a level I trauma center with standby thoracic surgery.
Overall, well-designed study, methodology reproducible but need to be improved in some basic information. Discussion should be improved, as well.
However, I have several suggestions and objections to improve the study:
- Please provide a full title for each abbreviation mentioned in the abstract. (IQR - should be stated at the place where this abbreviation is first used).
Authors answer: Thank you for your comment. It is of great importance to outline all the abbreviations in order to improve the readability of the manuscript.
Action taken: Abbreviations have been written out where they appear for the first time.
- Please provide clear inclusion / exclusion criteria in methodology.
Authors answer: We apologize for the lack of clarity.
Action taken:
- Lines 75-77: During the time interval of 12 years (2009-2020), all patients that received a chest tube in one level I trauma centre were included. They have been identified using the German Diagnosis Related Group-System.
- Lines 81-85: Excluded were patients without traumatic chest injury, pre-treatment for longer than 24h in other hospitals, penetrating chest injury, incomplete data and false coding. The data have then been screened for ISS and thoracic abbreviated injury scale (AIS). All patients with ISS<16 and thoracic AIS<3 have been excluded (Figure 1).
- Please provide more baseline characteristics of the study population such as weight, height, BMI, comorbities…
Authors answer: We agree that more detailed information on patient characteristics improves the quality of the manuscript. However, for the clinical question asked in this paper we feel that it is more important to be more specific on concomitant injuries. Hence, we added a table specifying injuries of the shoulder girdle as well as pelvic and abdominal injuries to the supplementary section.
Action taken: Lines 175-176: Concomitant injuries of the shoulder girdle as well as abdominal and pelvic injuries are listed in the supplementary table S1.
- Please provide exact p-values instead of ‘’p>0.05’’ or ‘’p<0.05’’ through the results section.
Authors answer: Thank you for your comment.
Action taken: We noted the specific p values throughout the manuscript.
- The authors stated that the predictive value for thoracotomy was calculated using a ROC curve mode. I do not see clear results of ROC analysis in the results section. Please can you provide area under the curve (AUC), sensitivity, specificity as well as ROC-curve diagram?
Authors answer: Thank you. We need to add more information on this point into the manuscript.
Action taken: Lines 151-153: The area under the curve was 0.991. Data on specificity and sensitivity as well as the ROC-curve diagram can be found within the supplements (Figure S2, Table S1).
- Discussion is poorly designed. Discussion section needs to be re-written/re-arranged. Do not present a review of literature in this section. Do not discuss your findings piecemeal. Focus on results from the main objectives of the study. Write in four sequential paragraphs (without headings); (i) summary (not data) of findings from present study; (ii) logical and coherent comparison with existing literature with focus of comparison on main objective(s); (iii) limitations of the study and (iv) Implications for practice/policy/research with a concluding statement.
Authors answer: Thank you for your honest opinion on the discussion section. We restructured the discussion all together. A limitation section has also been added.
Action taken: We restructured the discussion and added a limitations section.
- Limitations of the study should be clearly stated at the end of discussion.
Authors answer: Thank you.
Action taken: Lines 295-374: Limitations of this study include the small number of UT, the retrospective design and data collected in a single institution. Furthermore, within the study period VATS and rib plating have been used with increasing frequency in trauma surgery due to technological progress. Another limiting factor is the heterogeneity of injury mechanisms and injury patterns in polytraumatized patients which make it necessary to evaluate each patient individually. Thus, the measurement of the chest tube output can only be one information amongst others to make a clinical treatment decision.
Reviewer 2 Report
This is a good article to declare one of critical predictive factors for urgent thoracotomy after chest tube placement for blunt chest trauma.
However, there are some minor points to be refined as followings.
I wonder if urgent thoracotomy is the best option for active chest bleeding in blunt chest trauma in polytraumatized patients or not, although the author also mentioned it in Discussion (L199-203, P8)
- The author should show the patients who were treated by IVR and VATS instead of urgent thoracotomy to control intrathoracic bleeding in NT (n=225).
- I believe that contrast CT and following angiography (if necessary) are critical examination when we judge to treat or observe intrathoracic bleeding. Can the author justify that the chest tube output is the predictive factor for urgent thoracotomy, although the statement that pts with a chest tube output of more than 1300 ml within 24 h after treatment should be considered for a level I trauma center is persuasive (in conclusion)?
- Figure 3. (p6) The authors should add figure legends what do the symbols of ‘asterisk’ and ‘open circle’ in NT mean, and describe also the details in text.
Author Response
Reviewer 2:
This is a good article to declare one of critical predictive factors for urgent thoracotomy after chest tube placement for blunt chest trauma.
However, there are some minor points to be refined as followings.
I wonder if urgent thoracotomy is the best option for active chest bleeding in blunt chest trauma in polytraumatized patients or not, although the author also mentioned it in Discussion (L199-203, P8)
- The author should show the patients who were treated by IVR and VATS instead of urgent thoracotomy to control intrathoracic bleeding in NT (n=225).
Authors answer: Thank you for your important comment. There were no patients treated using IVR. Within the NT group, 3 patients received delayed thoracotomy (atypical wedge resection, because of air leakage (two times) and one purulent infection (lines 152-154)). For all of them, surgery was started with VATS and converted to open thoracotomy.
Action taken: Lines 151-153: For all of them, surgery was started with video-assisted thoracoscopic surgery (VATS) and converted to open thoracotomy.
- I believe that contrast CT and following angiography (if necessary) are critical examination when we judge to treat or observe intrathoracic bleeding. Can the author justify that the chest tube output is the predictive factor for urgent thoracotomy, although the statement that pts with a chest tube output of more than 1300 ml within 24 h after treatment should be considered for a level I trauma center is persuasive (in conclusion)?
Authors answer: We agree that contrast enhanced computer tomography gives important information on active intra-thoracic bleeding. Thus, according to the protocol of our institution all polytraumatized patients receive a contrast CT. However, within our study none of the patients received IVR. Hence, we can not make any conclusions on this point.
Action taken: None.
- Figure 3. (p6) The authors should add figure legends what do the symbols of ‘asterisk’ and ‘open circle’ in NT mean, and describe also the details in text.
Authors answer: Thank you. It is important to describe Fig. 3 in more detail.
Action taken: Lines 157-158: o indicates datapoints within 1.3-3 IQR, * above 3 IRQ.
Round 2
Reviewer 1 Report
The authors improved the manuscript but still they did not respond adequately to all what they were asked:
- The authors were asked to provide more baseline characteristics of the study population such as weight, height, BMI, comorbities… I do not see any of these characteristics in the revised manuscript. Please revise.
- Data on specificity and sensitivity as well as the ROC-curve diagram should be included in the manuscript, not supplementary material. Also, from the supplementary Table it is hard to see sensitivity and specificity. Please provide exact information in the line with AUC. Also include the ROC diagram in the main document and remove it from supplementary material.
Author Response
- The authors were asked to provide more baseline characteristics of the study population such as weight, height, BMI, comorbities… I do not see any of these characteristics in the revised manuscript. Please revise.
Response: I apologize for the misunderstanding. We now completed table 1 with age, sex, body weight, body height, BMI and comorbidities.
Action taken:
- Line 90: Table 1 was completed.
- Line 419: Acknowledgements: We thank Lucas Tizek for his help with additional data collection.
- Data on specificity and sensitivity as well as the ROC-curve diagram should be included in the manuscript, not supplementary material. Also, from the supplementary Table it is hard to see sensitivity and specificity. Please provide exact information in the line with AUC. Also include the ROC diagram in the main document and remove it from supplementary material.
Response: Thank you for your comment. We now added the ROC-Curve, sensitivity, specificity, AUC to the main document.
Action taken:
- Line 170: Figure 4 ROC curve was added.
- Line 183: Figure 4. ROC-curve diagram on chest tube output and need for UT shows a sensitivity of 100% and specificity of 95.3% at 1270 ml chest tube output with an Euclidean distance of 0.047. The area under the curve is 0.991.
- Line 164-165: Analysing the ROC-curve diagram the area under the curve was 0.991. A sensitivity of 100% and specificity of 95.3% at 1270 ml chest tube output with an Euclidean distance of 0.047 was calculated (Figure 4).